# Digital Mass Hysteria during Pandemic? A Study of Twitter Communication Patterns in the US during the Stages of COVID-19 Vaccination

**DOI:** 10.3390/bs14050389

**Published:** 2024-05-06

**Authors:** Dohyo Jeong, Jessi Hanson-DeFusco, Dohyeong Kim, Chang-Kil Lee

**Affiliations:** 1School of Economic, Political and Policy Sciences, University of Texas at Dallas, Richardson, TX 75080, USA; dohyo.jeong@utdallas.edu (D.J.); jessi.defusco@utdallas.edu (J.H.-D.); dohyeong.kim@utdallas.edu (D.K.); 2Department of Urban Policy and Administration, Incheon National University, Incheon 22012, Republic of Korea

**Keywords:** COVID-19, vaccination, sentiment analysis, interrupted time series, strategic action fields

## Abstract

This study examined the public’s sentiments about vaccines by analyzing Twitter data during the CDC’s vaccination management planning stage in the United States. Sentiment scores were assigned to each tweet using a sentiment dictionary and the sentiment changes were analyzed over 52 weeks from November 2020 to November 2021. An interrupted time series model was used to analyze the difference in sentiment, which revealed that there was a shift. Initially, overall sentiments were negative but became positive as the stage of general vaccine supply approached. However, negative sentiments sharply rose when the vaccine supply transitioned to the phase of universalization. The results identified two dominant strategic action fields for vaccines providing polarized messages on Twitter and the negative trend was strong for most of the period. The findings highlight the importance of managing strategic action fields on social networks to prevent mass hysteria during vaccine policy implementation. This study stresses the significance of effectively managing strategic action fields on social media platforms to prevent mass hysteria while implementing vaccine policies.

## 1. Introduction

In late 2020, the United States approved multiple COVID-19 vaccines (including Moderna and Pfizer-BioNTech) for emergency usage against the coronavirus disease 2019. Due to the constrained availability, the distribution of COVID-19 vaccinations in the United States was carried out gradually, prioritizing healthcare workers and vulnerable populations during the first phase. Subsequently, a national vaccination campaign plan was put into effect for the general population, with oversight provided by the Centers for Disease Control and Prevention (CDC) and other key government agencies [1,2]. During the vaccination process in the United States, however, several challenges arose that hindered the vaccination rate. Two major factors that hindered the vaccination rate in the United States were low trust in the government [3] and negative sentiments about vaccines spread on social media [4]. These have led to a plateau in vaccination in the United States since April 2021. It is true that the United States experienced a slower initial rollout of COVID-19 vaccinations compared to the United Kingdom, which began vaccinating at a similar time. In addition, some countries like South Korea and China, which started vaccinating later than the United States, have achieved higher vaccination rates in a shorter time frame.

As the distribution plan for COVID-19 vaccines in the United States entered different phases throughout the year, there was a noticeable increase in social media chatter debating various issues around the vaccine [5,6]. Vaccine hesitancy and skepticism about the vaccine’s safety and efficacy fueled these discussions, with many individuals expressing concerns and sharing misinformation or disinformation on social media platforms [6]. The launch of COVID-19 vaccines indeed provoked a group reaction not only among those receiving the new vaccines but also among those awaiting vaccination or considering whether to be vaccinated.

According to Clements (2003), the group reactions to COVID-19 vaccines could be classified as mass hysteria (MH) or mass psychogenic illness (MPI) [7]. This article proposes a new theoretical framework called digital mass hysteria (DMH), which explains how new-disease pandemics can cause social media platforms like Twitter to become vehicles for spreading mass hysteria. According to this framework, the shock of a new-disease pandemic can create chaos and uncertainty, leading people to seek meaning and understanding through social media. The article argues that DMH is more likely to occur in the context of new, contagious pandemics with limited prior experience or knowledge among government leaders, global health experts, and the public.

The research presented in this article aims to map the digital mass hysteria (DMH) surrounding COVID-19 vaccination chatter on Twitter from November 2020 to November 2021. The main goal is to provide insight into how this phenomenon occurred and, secondly, to offer guidance on how national governments and health agencies can better plan and manage DMH in the future. By analyzing the patterns and content of Twitter discussions around COVID-19 vaccines, this research seeks to identify key factors contributing to the spread of mass hysteria and offer effective communication strategies to counteract these effects.

## 2. Theoretical Background

### 2.1. US COVID-19 Vaccination Management Plan

During the pandemic, the primary objective of the government has been to ensure that there are enough COVID-19 vaccines available to all citizens who wish to be vaccinated [8]. Nevertheless, because the COVID-19 vaccination initiative is still in its initial stages, there could be constraints on vaccine availability. To address this issue, the US CDC released the government’s guidebook on vaccination management on 29 October 2020. One of the main challenges in the US vaccination management plan was that both the CDC and state governments had to limit the supply of vaccines at the beginning of the vaccination program. To ensure that the limited vaccine supply was utilized efficiently, the CDC divided the vaccine supply policy into three phases. In Phase 1, the initial phase of the COVID-19 vaccination, limited initial vaccine doses were distributed. By employing this distribution approach, the CDC aimed to enhance vaccine acceptance and safeguard public health while reducing wastage and inefficiencies. In the subsequent Phase 2, as the supply of vaccines grew, distribution broadened, enabling more of the population to access immunization services. Finally, in Phase 3, vaccine supply was sufficient to exceed demand [8]. This phased approach was designed to ensure that the vaccine was distributed equitably and efficiently, while also prioritizing the most vulnerable and at-risk populations during the initial stages of the vaccination program.

The CDC’s vaccine supply management guidelines recognize the importance of public trust and support for maintaining a stable vaccine supply. However, it is important to note that public sentiments toward vaccines can vary depending on the vaccination phase and each pandemic phase [9]. As a result, it is crucial to capture and manage these varying public sentiments throughout each phase of the vaccination management process, ensuring that public trust and support remain high, which is essential for maintaining an adequate and consistent vaccine supply.

### 2.2. Mass Hysteria and Sensemaking in Crises

In the face of a newly emergent pandemic, the amount of unknown information and uncertainties can rapidly increase, leading to widespread confusion, social unrest, and political-economic tensions [10,11,12]. During the 2014–2016 Ebola crisis, West African nations like Liberia experienced mass hysteria (MH). Public fear, anxiety, violent crimes, and assaults increased as the disease spiked in August–September 2014 [13,14]. During the Ebola crisis, MH not only affected West African countries but also spread digitally on a global scale [14]. Governments and international partners, including the World Health Organization (WHO), attempted to prevent the spread of the disease and raise awareness through public messaging efforts in early 2014 but their efforts were poorly coordinated and ineffective. As the Ebola crisis continued to escalate, misinformation, disinformation, and fearmongering began to spread, eroding public trust in national response efforts [15,16,17]. While Ebola was an epidemic, it was a transcontinental epidemic, meaning that as a health crisis, it took on the threat of becoming a pandemic in the eyes of many experts. Ebola-based policies informed 70% of all WHO national health policies and planning prior to COVID-19 [18]. Ebola and COVID-19 were thus interrelated and both transnational and transcontinental health crises, with misinformation and disinformation breaking out in relation to COVID-19.

Misinformation about the COVID-19 vaccine mainly comes from errors or misunderstandings of information [19,20]. For example, exaggerated or incorrect information about vaccine side effects and misunderstandings about vaccine ingredients [21] are often spread without the person conveying it knowing it is untrue. On the other hand, disinformation manipulates public opinion by intentionally creating and spreading false information [19]. The COVID-19 vaccine includes specific groups or individuals creating and spreading false information to intentionally denigrate the vaccine’s effectiveness or promote distrust of the vaccine [21]. In this way, as misinformation and disinformation about COVID-19 spread, these can lower the vaccination rate and lead to results that run counter to public health goals, ultimately leading to a mass hysteria against the COVID-19 vaccine, like Ebola cases [18].

Mass hysteria (MH) refers to a phenomenon marked by the simultaneous manifestation of comparable symptoms, anxieties, or phobias among a community of individuals united by a shared conviction [7,13,14,22]. There are five primary causes of MH: unsubstantiated but superficial symptoms, rapid onset and reversible symptoms that occur in isolated groups, anxiety-provoking situations, and symptoms transmitted through sight or hearing [23]. Symptoms of MH are often caused by stress and anxiety associated with a perceived threat, and MH can appear in various groups depending on the social environment or the cause of its spread [22,24]. Initial studies on COVID-19 have noted trends of MH related to the pandemic, particularly among younger populations [22] and with regional differences between rural and urban areas [24]. However, COVID-19 is only one of many recent emergent pandemics that have impacted the globe, and a comparison of recent pandemics can help identify trends in how pandemics affect our contemporary digital world.

One important consideration is that the public’s response to pandemic messaging is highly dependent on the credibility of the sources providing that information. Leaders and lead institutions can utilize sensemaking to find policy solutions and guide action [25,26]. Yet, the US struggled to take control of pandemic messaging during the first two years of COVID-19, leading to confusion and chaos. In times of crisis, sensemaking is the process by which people give meaning to collective experiences, whether positive or negative [27,28]. The concept of sensemaking is closely related to MH but they are not synonymous. While sensemaking is a necessary process for navigating complex and uncertain situations, MH is a phenomenon in which a group of people collectively experiences irrational and intense emotional reactions to a perceived threat. This paper posits that sensemaking can potentially lead to MH when the level of negativity escalates excessively. In this sense, MH can be seen as a byproduct of sensemaking gone wrong. This can occur when a group of people collectively experiences a high level of stress, anxiety, or fear due to a perceived threat and their sensemaking processes become distorted or overwhelmed by the intensity of their emotions.

### 2.3. Theoretical Framework

In this paper, we use the theoretical framework of digital mass hysteria (DMH) to recognize the role of digital and social media platforms in shaping public perception and sensemaking during times of crisis. DMH refers to the collective occurrence of irrational and intense emotional reactions to a perceived threat that is propagated and amplified through digital and social media platforms. DMH arises in a complex adaptive system (CAS), where multiple actors and factors interact in a nonlinear and dynamic way. The digital and social media platforms act as catalysts for the spread of DMH as they allow for the rapid dissemination of information, including misinformation and disinformation. As citizens try to make sense of the crisis, they may encounter conflicting information, leading to confusion, anxiety, and fear. Drawing on Fligstein and McAdam’s (2012) theory of fields and sensemaking, emergent pandemics such as Ebola and COVID-19 can be seen as exogenous shocks that can disrupt social systems, political stability, and economies. Such shocks trigger the emergence or modification of new strategic action fields (SAFs) as actors attempt to make sense of the situation and respond to it [29]. In this context, DMH can arise when the shock greatly disrupts the system to the extent that people turn to digital communication for sensemaking. However, the resulting digital chatter can be chaotic, especially when emerging or changing fields compete to dominate the narrative and disseminate conflicting and contentious information to the public.

## 3. Materials and Methods

The research unit for text and sentiment analysis focuses on tweets posted on Twitter. The analysis aims to compare the characteristics of positive and negative sentiments expressed in individual tweets. To collect data, the Twitter Academic API is used to randomly select 45,159 tweets spanning 52 weeks (Phase 1: 15 weeks, Phase 2: 24 weeks, Phase 3: 13 weeks), starting from 1 November 2020, shortly after the CDC released guidelines on vaccine management, and ending on 30 November 2021, when vaccines were widely available nationwide and free. To ensure the relevance and accuracy of the data collected, search parameters were set to include only English-language tweets and filtered to only tweets in which the user set their country to the United States. Also, to capture COVID-19 vaccination-specific discourse, the search keywords used were “vacc”, “vax”, “vaccine”, and “vaccination”. And “Moderna” and “Pfizer” were added as keywords to reflect mentioning a specific company as an alternative word to the “vaccine”. These filters were selected to broaden the scope of the data collected while concentrating on vaccination, capturing the diverse ways Twitter users discuss vaccines. The chosen timeline spans crucial phases of vaccine distribution and public acceptance, offering a comprehensive dataset for analyzing shifts in sentiment over time. The data used for the analysis is publicly available and can be accessed through the Harvard Dataverse (https://dataverse.harvard.edu/dataset.xhtml?persistentId=doi:10.7910/DVN/5DMXUO).

The quantitative analysis employs two primary empirical methods: sentiment analysis and interrupted time series analysis (see Figure 1). These two methods were utilized to examine the differences in sentiments expressed in tweets about vaccination, disaggregated across the three stages of the CDC’s vaccination management plan [8]. First, sentiment analysis was conducted by determining the sentiment of each tweet using the AFINN sentiment dictionary. This dictionary assigns numerical scores to words to determine the sentiment expressed in given texts. Each word is scored according to its expressive strength as defined in the AFINN dictionary. The sentiment score for individual tweets was computed by summing up the sentiment scores of all the words contained within each tweet. Subsequently, the weekly score was determined by averaging the scores of all tweets within that week. The intensity of negative or positive sentimental tweets for each week is calculated by estimating the value. This method assumes that the average of positive and negative sentiment provides a balanced representation of the overall emotional tone expressed in tweets during that period. Next, an interrupted time series model was used to analyze for differences in sentiments according to the vaccination stage of the CDC. This model was employed to estimate whether the proportion and intensity of negative or positive tweets for vaccination differ according to the management stage. The equation for the interrupted time series model is:$${Sentiment\;score}_{w}=\beta_{0}+\beta_{1}T_{w}+\beta_{2}{P2}_{w}+\beta_{3}{P2}_{W}T_{w}+\beta_{4}{P3}_{W}+\beta_{5}{P3}_{w}T_{W}+\epsilon_{w}$$

$\beta_{1}$—during Phase 1; $\beta_{2}$—the change immediately after from Phase 1 to Phase 2. $\beta_{3}$—difference between the slopes of Phase 1 and 2; $\beta_{4}$—change between Phase 2 to Phase 3; $\beta_{5}$—difference appearing overtime after changing to Phase 3; P2—Phase 2; P3—Phase 3.

${Sentiment\;score}_{w}$ is the aggregated sentiment score measured at each equally spaced week point “*w*”. $T_{w}$ is the week since the start of the vaccination, ${P2}_{w}$ and ${P3}_{w}$ are included as dummy variables to represent each vaccination phase, which allows us to estimate the effect of changes in each phase. These variables allow us to precisely measure the impact of moving from one stage to another while controlling for ongoing time trends within each stage [30]. For example, this ITSA approach helps address the spillover effects of sentiment from one phase (t − 1) to the next (t). By incorporating these step-by-step changes, ITSA allows us to discern whether observed transitions in sentiment are statistically attributable to the step transition itself or are part of a broader, ongoing trend [30]. It will also help us understand how sentiments change over time as vaccination management strategies change [30]. These two methods provide a comprehensive analysis of the sentiments expressed on Twitter about vaccination and offer insights into how these sentiments may vary over time and across different stages of vaccination management.

## 4. Results

### 4.1. Descriptive Statistics

Each tweet was assigned a sentiment score, with positive tweets having a score of 1 or higher, and negative tweets having a score of −1 or lower. Examples of tweet scoring are shown in Table 1. Weekly tweet scores were calculated by averaging the sentiment score for each of the 52 weeks analyzed. To compare the intensity of each sentiment, the absolute value of each sentiment score was taken. Sentiment scores of 0 were excluded from the analysis and only positive and negative sentiments were compared. A histogram (Figure 2) was used to visualize the distribution of positive and negative sentiment scores with zero points excluded.

The sentiment analysis revealed that the average sentiment score for positive tweets is 2.56, with the highest value being 15. On the other hand, the average sentiment score for negative tweets is 2.97, with the highest value at 24. The results suggest that negative sentiment scores were more intense compared to positive sentiment scores. This observation is supported by the trend line for sentiment scores, as shown in Figure 3.

In Figure 3, the most frequently occurring word at each point is marked, providing insight into the specific topics and concerns associated with the sentiment intensity. Notably, the intensity of negative vaccine tweets consistently outweighs that of positive tweets throughout all phases, with a brief spike in positive intensity during Phase 1. As the vaccine becomes more readily available during Phase 2, the intensity of negative tweets continues to escalate, potentially indicating public hesitancy and distrust leading to noncompliance with vaccination efforts. The interrupted time series analysis will further explore these patterns by estimating the statistical differences in sentiment changes over time.

### 4.2. Interrupted Time Series Analysis

To evaluate changes in sentiment intensity over time, this study conducted an interrupted time series analysis. The analysis was conducted in two parts: first, changes in overall sentiment intensity over time were evaluated and second, both negative and positive sentiment intensity were analyzed for significant differences in each stage. The analysis results are presented in Figure 4, which shows the interrupted time series for each phase of the overall sentiment score.

During Phase 1, the overall sentiment average shows a statistically significant positive coefficient (Coef. = 0.089, *p*-value < 0.001), indicating a robust increase in positive sentiment toward vaccination. This positive trend signifies public optimism during the early stages of vaccine rollout. However, the onset of Phase 2 marks a significant downturn, as evidenced by a negative coefficient for the overall sentiment at the start of this phase (Coef. = −0.163, *p*-value = 0.516). Although this coefficient is not statistically significant, the negative sentiment average significantly declines (Coef. = −0.552, *p*-value = 0.029), highlighting a marked increase in negative sentiments. This shift correlates to rising vaccine hesitancy and the spread of antivaccination messaging as the vaccine becomes more widely available. The coefficients for Phase 2 over time (Coef. = −0.104, *p*-value < 0.001) further confirm a continued decrease in positive sentiments, reflecting growing public concerns or dissatisfaction with the vaccination process. The significant negative trend for negative sentiments over time in Phase 2 (Coef. = −0.063, *p*-value = 0.006) suggests a gradual stabilization or recovery from initial negative reactions as the phase progresses. Transitioning into Phase 3, the coefficient for the start effect on overall sentiment (Coef. = 0.179, *p*-value = 0.235) and negative sentiment (Coef. = −0.150, *p*-value = 0.296) indicates an attempt at sentiment recovery, although these changes are not statistically significant, suggesting variability in public reception. The data further reveal that while negative sentiments are significantly influential, positive sentiments during these phases do not show statistically significant changes, as indicated by the lack of substantial coefficients for positive sentiment over time in Phase 2 and Phase 3.

As shown in Table 2, the statistical analysis shows that negative sentiments toward vaccination are significant, while positive sentiments are not, indicating the significant influence of mass hysteria as a byproduct of the sensemaking in crises for the COVID-19 vaccine. The trendline and discontinuous time series analysis suggest a unique point for each stage of vaccine management, with a notable difference between Phase 1 and Phase 2 when full-scale vaccination begins in the US. During this period, there is a drastic shift in tweets as antivaccination messaging grows, leading to a significant increase in negative sentiment intensity. The results suggest that effective communication and messaging strategies are crucial for mitigating vaccine hesitancy and increasing public trust in vaccination during times of crisis.

## 5. Discussion

The concept of sensemaking can be useful for leaders to gain a better understanding of their environment and, in turn, engage in effective leadership activities such as creating a vision, building relationships, and fostering innovation [25]. However, about the COVID-19 vaccine, digital mass hysteria (DMH) may occur due to the unpredictability of the vaccine, intolerance of uncertainty, and perception of the risk of infection. Vaccine-targeted DMH can cause negative psychological reactions, including behavioral changes, emotional distress, and public avoidance [5,12].

Once DMH becomes established in a population, addressing it becomes a significant challenge. Due to the advancement of media and the substantial influence of social platforms, DMH can spread quickly and widely, making containment difficult. The problem is compounded by the fact that misinformation and disinformation can spread rapidly on social media, regardless of their veracity [19,20]. The impact of social media on pandemics and vaccine management can be catastrophic for public health [5,6,31]. For instance, when news of the first deaths during the initial vaccine rollout and an uptick in new COVID-19 cases hit the media, this may have exacerbated people’s fears, anxiety, helplessness, and frustration about the vaccine [4,32]. This, in turn, can fuel collective fear and hysteria among anxious individuals, leading to conflicts between health authorities and the public [12].

Figure 5 depicts the structure of digital mass hysteria during the COVID-19 pandemic based on the results of the Twitter analysis conducted in this study. As shown in the figure, this study identifies two specific fields that dominated public discourse and sensemaking around the pandemic response during the American COVID-19 response. Representative words that emerged from positive sentiments included “care”, “help”, “free”, “protect”, and “proof”. Conversely, the main words associated with negative sentiments were “risk”, “wrong”, “sad”, “Trump”, and “mandatory”. These fields are characterized by group identities that fuel opposing maps related to COVID-19 vaccination [14,33,34]. Different messaging is viewed as credible by one side but often dismissed as disinformation or misinformation by the other, with much of this messaging propagated over social media platforms like Twitter [35]. This polarized discourse acted as a major catalyst for DMH starting in late 2020. For example, antivaccine sentiment often stems from concerns about the rapid development of vaccines and their possible side effects, which occurred by misinformation and disinformation spread through social media [19,33]. On the other hand, vaccination advocates focus on the safety and efficacy of vaccines approved by health authorities such as the CDC [32]. On one side of the discourse were right-wing politicians like President Donald Trump, the Republic Party, and his Response Team, often clashing with incumbents like the CDC, Democratic candidate Joe Biden, and the Left [32]. Antivaccination groups like ICAN increasingly blended with the Far-Right, especially as national elections approached [33,34].

This gap in views and opinions is amplified by digital mass hysteria. Within each distinct field, key messaging was disseminated to the public, mainly through social media, and often fueled divergent, highly politicized information about COVID-19 prevention and vaccination [36,37,38]. Each group interprets information based on rigid biases, polarizing public discourse, and complicating efforts to achieve widespread vaccine acceptance [33].

Twitter played a critical role during the COVID-19 pandemic as one of the major communication platforms utilized by both key field actors and the public. It provided a platform to make sense of the unknowns, speculations, and unverified facts surrounding the pandemic [35,36,37,39,40]. As the pandemic progressed, Twitter became a hub for conversations regarding the COVID-19 vaccination, with both positive and negative messaging circulating at different stages of the US government vaccination plan. This chatter helped establish what Fligstein and McAdam refer to as “signals and tags” among main strategic action fields and public awareness of whether and how to get vaccinated, much of which was highly politicized and polarizing [29] (pp. 3–31). This polarized discourse led to DMH.

Prior large-scale pandemics can offer invaluable lessons about monitoring digital and media communication trends and can identify harmful negative messaging and offer solutions to combat DMH. During the summer and early fall of 2014, growing MH during the Liberian Ebola crisis likely contributed to disease spread and ineffectual policy mandate compliance. But, in November 2015, international and national stakeholders volitionally partnered to enact a communications plan using radio, posters, television, and social media to better combat negative messaging and to promote positive messages. This strategy eventually contributed to rebuilding public trust and streamlined collective action using a hybrid policy implementation approach to combat Ebola [15,16,17]. Such efforts indicate how coordinated communications over different media platforms can help create order and meaning out of chaotic events like pandemics [18].

## 6. Conclusions

This case study provides valuable insights into how governments and health agencies at both national and international levels can collaborate to combat negative sentiments on digital platforms like Twitter. Instead, they can promote positive and accurate messages about vaccination and disease prevention [4]. Through data analysis of average weekly tweet scores, it is evident that negative sentiments consistently dominated Twitter throughout November 2020–2021. While there was a brief spike in positive messaging during Phase 1, as mass inoculation preparation approached, this appears to be short-lived. The timing of this positive spike indicates a sense of public relief and anticipation to be vaccinated. However, this positive sentiment quickly faded as a growing tide of negative vaccination tweets emerged in Phase 2 and even Phase 3.

The shock of COVID-19 led to the formation of two dominant SAFs that were mainly distinct but sometimes overlapping. The first SAF consisted of individuals who identified with antivaccination movements, held far-right or Republican political values, opposed government interference and pandemic mandates, and supported free-market economic principles. The second SAF primarily included provaccination groups, left-wing politicians and supporters, and organizations such as the CDC and presidential candidate Joe Biden. Key messaging was disseminated within each distinct field, often through social media, resulting in divergent, highly politicized information about COVID-19 prevention and vaccination. This polarized messaging helped to establish signals and tags among main strategic action fields and public awareness of whether and how to get vaccinated, with much of it being highly politicized and polarizing [36,37,38].

While this study provided valuable insights into the emotional changes and digital mass hysteria during a pandemic, it is essential to note its limitations. The 45,159 tweets used in this study may represent a relatively limited amount of data to reflect the opinions of the entire US populace fully. However, these tweets were collected based on the capabilities and constraints of the Twitter Academic API and were filtered and keyword-targeted to meet the objectives of this research. This sample was gathered to capture a diverse range of opinions and sentiments about COVID-19 vaccinations and reflects sentiments about the vaccine within the limitations of available data. It demonstrates that even a small subset of social media data, when appropriately collected and analyzed, can effectively capture public opinion and trends [41,42].

Additionally, individuals’ tweets can vary significantly depending on the characteristics of the accounts they are following. Particularly, tweets from accounts with varying levels of popularity and credibility can significantly influence the sentiment expressed. However, in this paper, tweets collected via the Twitter API were treated equally without any weighting, as only user identifiers were available and there was no detailed information about the users’ account characteristics. Therefore, caution should be exercised when interpreting the results of this study. Future research may address these limitations by examining digital mass hysteria patterns in more detail, including specific time points and user characteristics.

## Figures and Tables

**Figure 1 behavsci-14-00389-f001:**
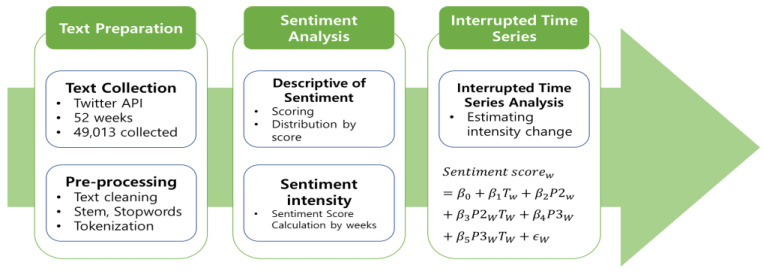
Data analysis framework.

**Figure 2 behavsci-14-00389-f002:**
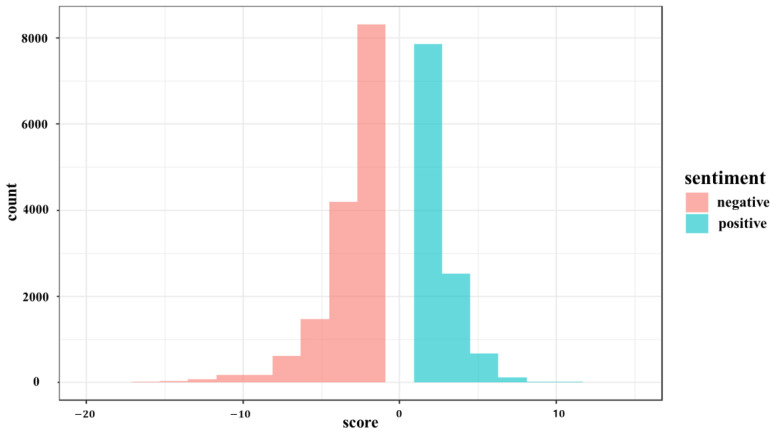
Histogram by sentiment score.

**Figure 3 behavsci-14-00389-f003:**
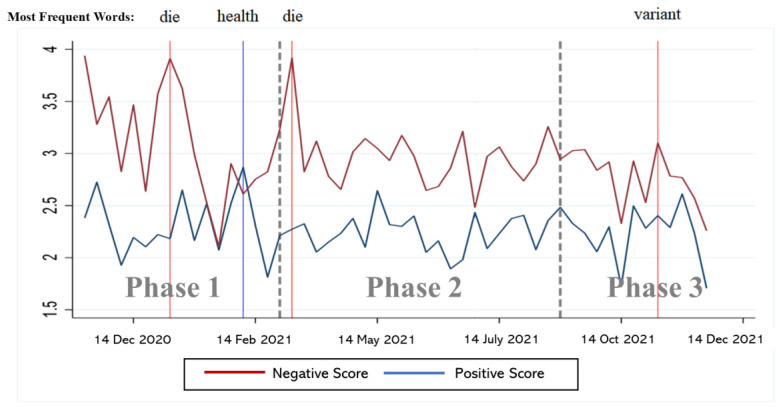
Trend line for the intensity of sentiment for US COVID-19 vaccine tweets.

**Figure 4 behavsci-14-00389-f004:**
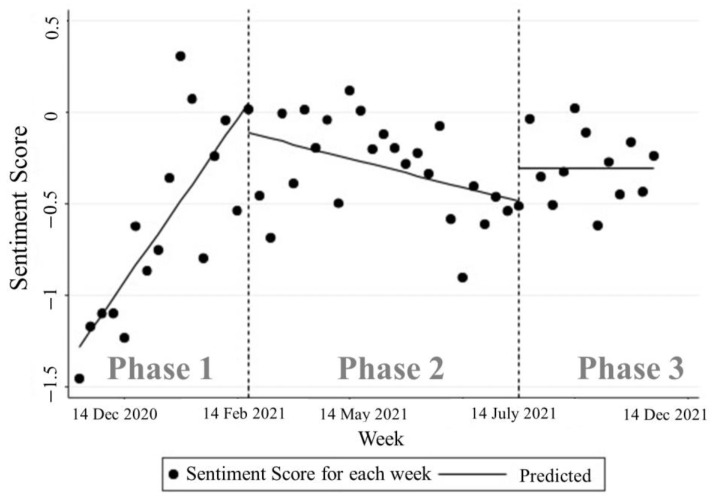
Interrupted time series analysis of weekly sentiment score.

**Figure 5 behavsci-14-00389-f005:**
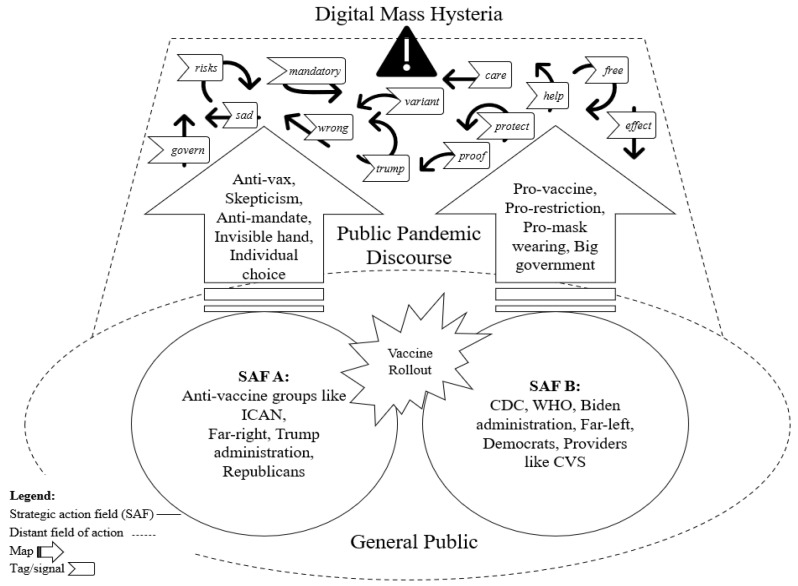
Structure of digital mass hysteria during the COVID-19 pandemic.

**Table 1 behavsci-14-00389-t001:** Examples of tweet scoring.

Tweets before Preprocessing	After Preprocessing	Score
There was a lot of hope and optimism in the library-turned-vaccination-suite. We were required to stay 15–30 min after the shot for monitoring and it was nice to just talk and laugh with others to feel like we were sharing a small victory. It’s been in short supply.	lot hope optim library turned vaccination suit requir stay 1530 minut shot monitor nice just talk laugh other feel like share small victori short suppli	|9|
RT @pettapallath @POTUS @VP Thanks for taking steps to make vaccination available for everyone before the target date. It would be great i??	thank take step make vaccin avail everyon target date great	|5|
RT @WhiteHouse\I will get vaccinated as soon as possible and I urge you to do the same. We need to protect our vulnerable neighbors rebu…	get vaccin soon possibl urg need protect vulner neighbor rebu	|1|
2 vaccination sites shut down in #ScarbTO with approximately 10,000 appointments cancelled. Where? the outrage by our #onpoli #TOpoli elected officials for our communities? Playing politics with our lives &amp; staying silent while health inequities continue to grow?! We see you.	2 vaccin site shut approxim 10,000 appoint cancel outrag elect offici communiti play polit live stay silent health inequ continu grow see	|−1|
@BjStov Me neither not that I think there is anything wrong with it. I just don’t know what the long term ramifications of the vaccine. Every year the flu vac is usually wrong. imho its not been tested enough.	neither think anyth wrong just know long term ramif vaccin everi year flu vac usual wrong imho test enough	|−6|
@Nation985 @Mr_Grant_I @notagain_ohno @NBCNews In the real world people will become ill and die from the vaccine. That’s a fact. WTF is wrong with you people?	real world peopl becom ill die vaccin fact wtf wrong people	|−11|

**Table 2 behavsci-14-00389-t002:** Interrupted time series analysis of sentiment intensity.

Average Sentiments	Overall Sentiment Average		Negative Sentiment Average		Positive Sentiment Average	
	Coef.	t	Coef.	t	Coef.	t
Phase 1	0.089 (0.018)	4.74 ***	0.073 (0.019)	3.72 ***	0.012 (0.017)	0.72
Phase 2 start effect	−0.163 (0.249)	−0.66	−0.552 (0.244)	−2.26 **	−0.265 (0.168)	−1.58
Phase 2 over time	−0.104 (0.020)	−5.12 ***	−0.063 (0.022)	−2.87 ***	−0.008 (0.017)	−0.47
Phase 3 start effect	0.179 (0.148)	1.20	−0.150 (0.142)	−1.06	0.0427 (0.133)	0.32
Phase 3 over time	0.016 (0.015)	1.03	0.029 (0.016)	1.78 *	−0.014 (0.021)	−0.71
intercept	−1.283 (0.097)	−13.10	3.625 (0.185)	19.54	2.257 (0.138)	16.31
Observation	52		52		52	
F (5, 46)	18.81 (Prob > F = 0.000)		5.57 (Prob > F = 0.000)		0.77 (Prob > F = 0.574)	
R^2^	0.5285		0.3444		0.0802	
rho	0.0210		0.0083		−0.0323	
Durbin-Watson	1.9865		1.9652		1.9210	

*p* < 0.1 *, *p* < 0.05 **, *p* < 0.01 ***.

## Data Availability

The data in this study are based on Twitter data for academic research, which is provided by the Twitter Developer Platform at https://developer.twitter.com/en/use-cases/do-research/academic-research.

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
