# Peer review of "Digital Mass Hysteria during Pandemic? A Study of Twitter Communication Patterns in the US during the Stages of COVID-19 Vaccination"

_behavsci, 2024, doi:10.3390/bs14050389_

Round 1

Reviewer 1 Report

Comments and Suggestions for Authors

This paper aims to study the phenomenon of Digital Mass Hysteria (DMH) on Twitter during November 2020 to November 2021. The data from Twitter was quantified using an NLP technique—sentiment analysis—with the AFINN sentiment dictionary. The authors employed interrupted time series analysis to observe changes in sentiment over different intervals of time. The results claim that DMH was present during the mentioned time.

The paper presents a very interesting concept and uses an appropriate statistical technique to support its claims. However, there are several areas of ambiguity and concerns regarding the dataset that need to be addressed:

Major Concerns:

  1. Dataset Size and Detail: The dataset, as described by the authors, comprises 45,159 tweets over 52 weeks, averaging around 900 tweets per week. This sample size seems too small to be representative of the broader population. The authors must provide more details about the dataset and justify why 900 tweets weekly accurately represent the larger population.
  2. Data Collection Specificity: The details of data collection are not sufficiently outlined. It's crucial for the authors to ensure that the tweets analyzed were specifically related to the USA. Since English tweets could originate from any geographical location and might not pertain to the USA, the methodology for collecting geographically relevant tweets needs to be clearly explained. For instance, an English tweet from the UK about vaccination issues should not be included if it does not relate to the USA. Unless this data collection process is clarified, references to the "US" should be reconsidered.
  3. Assumption of Linear Interaction Between Sentiments:
    • The justification for averaging out positive and negative sentiment scores is not provided. The authors should explain why, for instance, a positive sentiment score of 2 and a negative one of -1 would result in an average score of 0.5 for an individual.
      • It is acceptable if the authors assumed it, but the authors must mention it explicitly.
    • The unweighted averaging of tweets from accounts of varying popularity and credibility needs justification. The influence of tweets from more popular and credible accounts may significantly differ, and the authors should address why these tweets are averaged equally with those from less influential accounts.
  4. Lack of Consideration for Spillover Effects: Sentiment from a previous phase (t-1) may carry over to the next phase (t) to some extent. The authors should justify the model's exclusion of such spillover effects.
  5. Clarity in Results Interpretation: The interpretation of results, especially the magnitude of coefficients, requires further clarification for a comprehensive understanding.

Minor Concerns:

  1. Phase Timeframes: The timeframes for each phase are not clearly described. This information should be explicitly stated.
  2. Missing F-test P-value: The p-value for the F-test is missing in Table 2 and should be included for completeness.

Author Response

Our responses to the reviews are attached.

Reviewer 2 Report

Comments and Suggestions for Authors

Remarks

Although the theoretical framework is quite correct, there are concepts that need to be defined and differentiated, such as disinformation and misinformation, or pandemic and epidemic. the latter concept would include Ebola, which is referred to in the research as a pandemic. Both terms, from a communicative point of view, would be encompassed in a more general idea, that of health crisis.

These ideas are differentiated in some articles such as:

Aïmeur, E., Amri, S., & Brassard, G. (2023). Fake news, disinformation and misinformation in social media: a review. Social Network Analysis and Mining13(1), 30.

Pérez Escolar, M., Lilleker, D., & Tapia Frade, A. J. (2023). A systematic literature review of the phenomenon of disinformation and misinformation.

Also, the term vaccination is a broad concept of which specific trademarks are part. Are the identified trends contrary and negative sentiments to all of them, or are there variants, is there also an evolution, and have the plans developed to improve the acceptance of vaccination had a global sense or catered to these hypothetical differences?

On the other hand, in order to enrich the text, the arguments put forward by anti-vaccine sentiments in support of such approaches and those used by advocates could be provided, so that the analysis not only contrasts feelings and emotions towards vaccination but also logical reasons, which often underlie emotions and lead the audience to ideological polarisation.

Finally, it would be illustrative to include key words defining sentiments for and against vaccination in the different periods, including representative tweets, so that the academic nature of the text shows illustrative examples.

Author Response

(The authors gave the same response as above.)

Round 2

Reviewer 1 Report

Comments and Suggestions for Authors

Thank you for your responses; you have addressed my comments very well.

However, there is a small but significant oversight that cannot be overlooked. Unfortunately, I must ask for another revision.

Minor Revision:

  • The p-values associated with the regression results were not reported. Generally, coefficient reporting is accompanied by p-values, not t-values. Therefore, the authors must revise this to include the p-values: (coef. = xxx, p-value = 0.xxx), and it should be p-value<0.001 if the p-value is less than 0.001.

Author Response

We have received minor comments on our revised manuscript entitled “Digital Mass Hysteria during Pandemic? A Study of Twitter Communication Patterns in the US during the Stages of COVID-19 Vaccination” Although minor, the comment was very helpful in further refining and substantially improving the manuscript. Following the suggestions, we have added the p-value information to coefficients as necessary. Along with this letter, you will find our revised manuscript with all the changes highlighted in yellow. Once again, we thank the reviewer for the careful review and consideration of our manuscript. 
